# Influence of Surface Treatments on Urea Detection Using Si Electrolyte-Gated Transistors with Different Gate Electrodes

**DOI:** 10.3390/mi15050621

**Published:** 2024-05-05

**Authors:** Wonyeong Choi, Seonghwan Shin, Jeonghyeon Do, Jongmin Son, Kihyun Kim, Jeong-Soo Lee

**Affiliations:** 1Department of Electrical Engineering, Pohang University of Science and Technology (POSTECH), Pohang 37673, Republic of Korea; pathfinder@postech.ac.kr (W.C.); ssh3290a@postech.ac.kr (S.S.); toaru124@postech.ac.kr (J.D.); jmson@postech.ac.kr (J.S.); 2Division of Electronics Engineering, Jeonbuk National University, Jeonju 54896, Republic of Korea; kihyun.kim@jbnu.ac.kr; 3Innovative General Electronic Sensor Technology (i-GEST) Co., Ltd., Pohang 37673, Republic of Korea

**Keywords:** biologically active field-effect transistor, electrolyte-gated transistor, enzyme sensor, pKa, urea detection

## Abstract

We investigated the impact of surface treatments on Si-based electrolyte-gated transistors (EGTs) for detecting urea. Three types of EGTs were fabricated with distinct gate electrodes (Ag, Au, Pt) using a top-down method. These EGTs exhibited exceptional intrinsic electrical properties, including a low subthreshold swing of 80 mV/dec, a high on/off current ratio of 10^6^, and negligible hysteresis. Three surface treatment methods ((3-amino-propyl) triethoxysilane (APTES) and glutaraldehyde (GA), 11-mercaptoundecanoic acid (11-MUA), 3-mercaptopropionic acid (3-MPA)) were individually applied to the EGTs with different gate electrodes (Ag, Au, Pt). Gold nanoparticle binding tests were performed to validate the surface functionalization. We compared their detection performance of urea and found that APTES and GA exhibited the most superior detection characteristics, followed by 11-MUA and 3-MPA, regardless of the gate metal. APTES and GA, with the highest pKa among the three surface treatment methods, did not compromise the activity of urease, making it the most suitable surface treatment method for urea sensing.

## 1. Introduction

Field-effect transistor (FET) biosensors have emerged as one of the most popular electric biosensors, renowned for their exceptional sensitivity, rapid response times, capacity for miniaturization, and cost efficiency [1,2]. These benefits stem from their operational principle, which relies on the gating effect of biochemical recognition on the sensing surface. Among FET sensors, electrolyte-gated transistors (EGTs) hold a leading position due to their high sensitivity [3,4,5,6,7]. EGTs utilize the gate electrode as the sensing surface, offering a broad sensing area and a high density of receptors, thereby achieving enhanced sensitivity.

EGTs have the potential to detect a wide range of analytes, depending on the receptors immobilized on the gate electrode. These analytes include pH, which is crucial in biomedical and environmental monitoring due to its impact on cellular function and ecological health [8]. Antigens are another key target, particularly in immunodiagnostic applications for detecting diseases such as COVID-19 [9]. DNA detection is essential in genetic analysis and diagnostics, aiding in disease identification and personalized medicine [10,11,12]. Enzyme sensors, in particular, have garnered significant attention for their various implications, such as medical diagnostics, food safety, and environmental monitoring [13,14,15].

In the field of enzyme sensors, enzymes serve as the bio-recognition elements, enabling the selective detection of target analytes. The immobilization of enzymes onto the sensing surface is a crucial step, as the immobilization process influences critical aspects such as the accommodation of bio-recognition elements, preservation of their activity, and resistance against nonspecific interactions [16,17]. Therefore, in EGTs where the gate electrode is used as the sensing surface, the selection of suitable surface treatments and gate electrodes plays a pivotal role. Surprisingly, there have been few studies on the optimal combinations of surface treatments and gate electrodes, leaving an ideal material or immobilization method for biological sensing largely undefined.

To address this gap, we conducted a comparative study on surface treatment methods and gate electrodes for detecting urea. Urea is a key indicator of renal function in medical diagnostics, with abnormal levels indicating potential health issues [18]. We applied three surface treatment methods ((3-amino-propyl) triethoxysilane (APTES) and glutaraldehyde (GA), 11-mercaptoundecanoic acid (11-MUA), 3-mercaptopropionic acid (3-MPA)) to EGTs with gate electrodes made of three different types of metals (Ag, Au, Pt), and validated the surface functionalization using gold nanoparticles (AuNPs). Subsequently, we compared the detection performance to identify the most effective immobilization method and gate electrode.

## 2. Materials and Methods

### 2.1. Material Preparation

Urease from Jack Beans (Type III, powder, 20,000 units/g), urea (molecular biology grade, powder), phosphate-buffered saline (PBS, pH 7.4), (3-amino-propyl) triethoxysilane (APTES, 99%), anhydrous ethanol (200 proof, 99.5%), glutaraldehyde (GA, 50%), 11-mercaptoundecanoic acid (11-MUA, 98%), 3-mercaptopropionic acid (3-MPA, 99%), 1-ethyl-3-(3-dimethylaminopropyl)carbodiimide (EDAC, 98%), N-hydroxysuccinimide (NHS, 98%), IgG from mouse serum (95%), anti-mouse IgG conjugated with AuNP, glucose, ascorbic acid (AA), and KCl were purchased from Sigma-Aldrich (Burlington, VT, USA). Urea and other biomolecules, such as glucose, AA, and KCl, were dissolved in the 1 × PBS solution. We denoted the concentration of urea with pUrea (pUrea = $-\log_{10}\left[\mathrm{Urea}\right]$).

### 2.2. Fabrication of EGTs

Using a top-down approach, the EGTs were fabricated on a silicon-on-insulator wafer (p-type, 10 Ω∙cm (100)) (Figure 1). Initially, a 140 nm-thick top silicon layer was thinned to 100 nm through thermal oxidation. The active regions (source, drain, channel) were defined using a KrF scanner and an inductively coupled plasma reactive-ion etching (ICP-RIE) process. To minimize etching-induced damage and serve as a blocking layer for the ion implantation process, a 20 nm sacrificial oxide was grown using a thermal furnace. Arsenic ions at a dose of 5 × 10^15^/cm^2^ and an energy of 60 keV were implanted into the source and drain regions, followed by rapid thermal annealing (RTA) at 1000 °C for 20 s. Subsequently, a 5 nm gate oxide layer was grown using thermal oxidation. The gate electrode and source/drain transmission lines were formed using a lift-off process with 500 nm of Ag, Au, or Pt. Finally, a 2 μm-thick SU-8 passivation layer was applied for electrical isolation, except for the gate sensing area, channel, and contact pads. The passivation layer prevents the current flow from the source and drain regions to the aqueous solution and gate region during measurement.

### 2.3. Electrical Characterization

Utilizing a semiconductor parameter analyzer (Keithley 4200, Keithley, Solon, OH, USA), the electrical characteristics were measured. The gate voltage (*V_G_*) was applied through a buffer solution, and the drain current (*I_D_*) was measured at a constant drain voltage (*V_D_*) of 0.1 V, with source and body voltages (*V_S_* and *V_B_*) set to 0 V. The *I_D_*–*V_G_* characterizations were conducted after a 10 min exposure of the 20 μL target solution. *I_D_* was restricted to 10^−7^ A to prevent device degradation during the characterization.

### 2.4. Functionalization of EGTs

Three distinct materials were employed for the immobilization of urease on EGTs, as listed in Table 1. For the surface treatment using APTES and GA, a UV/ozone treatment with an intensity of 200 µW/cm^2^ for 90 s was initially conducted on the device. After exposure to APTES at 55 °C for 1 min, the device was rinsed with anhydrous ethanol and immersed in a 2.5% GA solution (1 × PBS, pH 7.4) for 90 min. Then, it was washed with 1 × PBS and DI water (DIW) and dried using N_2_ blowing. The ethoxy group (OCH_2_CH_3_–) of APTES bound onto the gate electrode, and the remaining amine group (NH_2_–) coupled with the aldehyde group (CHO–) of GA, which can bind to urease. Conjugate length of APTES and GA was 1.5 nm [19,20]. The pKa values of APTES and GA are both 10 [21,22], which indicates minimal impact on the pH of the buffer.

11-MUA is a widely-used substance for metal binding [23,24,25]. The thiol group (SH–) of 11-MUA binds to the gate metal, and the remaining carboxyl group (COOH–) attaches to urease. The length of 11-MUA is 1.6 nm [26], and its pKa is 6.5 [27], which is lower than that of APTES and GA. This lower pKa is attributed to the low affinity of COOH– for hydrogen ions (H^+^), facilitating easy dissociation. 

The final material employed was 3-MPA, which binds to metals through the thiol group (SH–) and attaches to urease via the carboxyl group (COOH–) in the same manner as 11-MUA. The length of 3-MPA is 0.6 nm [28], the shortest among the three surface modifications, and its pKa of 5.2 [29] is even lower than that of 11-MUA.

The surface treatments for 11-MUA and 3-MPA were similar, as follows. Initially, the devices were immersed in a 20 mM solution of 11-MUA or 3-MPA in an ethanol base for 18 h. Then, the devices were rinsed with ethanol and DIW and dried with N_2_ blowing. Subsequently, the devices were incubated in a solution containing 20 mM EDAC or 20 mM NHS in a 1 × PBS base for 1 h to activate 11-MUA or 3-MPA, respectively. Finally, the devices were rinsed with 1 × PBS and DIW and dried with N_2_ blowing. After the surface treatments described above, a urease solution (10 mg/mL, 1 × PBS, pH 7.4) was applied to those devices for 18 h at 4 °C in a humid environment, followed by rinsing with 1 × PBS and DIW and drying with N_2_ blowing.

AuNP binding tests were performed to evaluate the surface functionalization for different gate materials. A solution of mouse IgG (1 mg/mL, 1 × PBS) was applied to the devices in a humid environment at 4 °C for 18 h, rinsed with 1 × PBS and DIW, and dried with N_2_ blowing. Then, a solution of anti-mouse IgG conjugated with AuNP of 10 nm diameter was applied to the metal for 30 min, rinsed, and dried. The binding of AuNPs was examined using scanning electron microscopy (SEM). As shown in Figure 2, the average densities of AuNPs were in the ranges of 450~550 µm^2^ for different surface treatments and gate metals. Each data point corresponds to the average measurement obtained from five samples.

## 3. Results and Discussion

### 3.1. Intrinsic Electrical Characteristics

Figure 3 shows the intrinsic transfer curve (*I_D_* vs. *V_G_*) for EGTs with different gate metals. Regardless of the gate metal, these devices exhibit outstanding characteristics, including a low subthreshold swing (*SS*) of 80 mV/dec, high on/off current ratio of 10^6^, and negligible hysteresis of less than 10 mV. The threshold voltage (*V_TH_*) is characterized as low as 0.55 V for Ag, 0.65 V for Au, and 0.7 V for Pt, respectively, which is mainly attributed to the metal work–function difference [30].

### 3.2. Urea Sensing Characteristics

Figure 4 shows the real-time responses for urea with urease-functionalized EGTs for 600 s. Compared with the negative samples (1 × PBS without urea, half-empty symbols), those exposed to a urea solution (pUrea 0.5, filled symbols) decrease monotonously and then saturate.

The voltage-related response (*R_V_*) and the current-related response (*R_I_*) are defined as follows [15,31,32]:(1)$$R_{V}=V_{G1}-V_{G0},$$
(2)$$R_{I}=\frac{I_{D0}-I_{D1}}{I_{D1}},$$
where *V_G_*_0_ and *V_G_*_1_ represent gate voltages at a fixed *I_D_*_0_ of 3 nA before and after the reaction, respectively. *I_D_*_0_ and *I_D_*_1_ represent drain currents at a fixed *V_G_*_0_ of 0.3 V before and after the reaction, respectively. 

Figure 5 illustrates the correlation between *R_V_* and pUrea based on three surface treatment methods (APTES and GA, 11-MUA, 3-MPA) for EGTs with three gate metals ((a) Ag, (b) Au, (c) Pt). Each point represents the average of five different devices. In all conditions, it can be observed that as the urea concentration increases, the signal amplifies until reaching saturation. Regardless of the gate metal, a similar trend of higher *R_V_* values in the order of APTES and GA, 11-MUA, and 3-MPA was observed.

Figure 6 presents the extracted metrics from the *R_V_* vs. pUrea results of Figure 5. Figure 6a represents the *R_V,max_*, which is defined as the maximum value from the fitted curve. Figure 6b shows the urea sensitivity related to *R_V_* (*S_V_*), determined by the slope of the logistic-fitted line of *R_V_* [15]. Figure 6c depicts the limit of detection (LOD) of *R_V_*, defined using the 3σ method from the logistic trend line [33]. It can be observed across all metrics (*R_V,max_*, *S_V_*, LOD) that APTES and GA exhibits the most superior characteristics, followed by 11-MUA and 3-MPA, regardless of the metal type.

Figure 7 illustrates the correlation between *R_I_* and pUrea based on three surface treatment methods (APTES and GA, 11-MUA, 3-MPA) for EGTs with three gate metals ((a) Ag, (b) Au, (c) Pt). Each point represents the average of five different devices. Regardless of the gate metal, a similar trend of higher *R_I_* values in the order of APTES and GA, 11-MUA, and 3-MPA was observed.

Figure 8 presents the extracted metrics from the *R_I_* vs. pUrea results of Figure 7. Figure 8a, Figure 8b, and Figure 8c depict *R_I,max_*, urea sensitivity related to *R_I_* (*S_I_*) and LOD, respectively, based on the gate metal and surface treatment method. Consistent with the extraction based on *R_V_*, it can be observed across all metrics (*R_I,max_*, *S_I_*, LOD) that regardless of the metal type, APTES and GA exhibits the most superior characteristics, followed by 11-MUA and 3-MPA.

Tendency observed among surface treatment materials stems from the relationship between their respective pKa values and urease activity. The activity of all enzymes, including urease, varies with pH. The jack bean urease used in this study exhibits maximum activity at pH 7.5, decreasing as the pH decreases [34]. Both 11-MUA (pKa 6.5) and 3-MPA (pKa 5.2) possess carboxy terminals (COOH–), making them prone to the easy decomposition of H^+^ ions, thus acidifying the surrounding solution (Figure 9). This acidity diminishes the activity of urease, consequently reducing the signal magnitude [35]. In contrast, APTES and GA (pKa 10) feature amine terminals (NH_2_) that resist the release of H^+^ ions, thus having minimal influence on the solution’s pH. Consequently, using APTES and GA does not compromise urease activity, making it the most suitable surface treatment method for urea sensing.

### 3.3. Selectivity Test

To demonstrate the specific detection of urea, a selectivity test was conducted (Figure 10). Each data point corresponds to the average measurement obtained from five devices. The immobilization of urease was carried out using APTES and GA, which was identified as the most effective surface treatment method. For reacting substances commonly found in blood vessels, such as glucose, ascorbic acid, and KCl, only *R_V_* values smaller than 10 mV were observed regardless of gate metals (Ag, Au, Pt). Additionally, when a high concentration (316 mM) of urea was reacted with the EGTs without urease immobilization, the resulting *R_V_* was less than one-third of the value observed when reacting with a 100 times lower concentration (3.16 mM) of urea on the EGTs with urease immobilization. This confirmed the successful immobilization of urease on the EGTs and reiterated the specific detection of urea through the reaction between urease and urea using all gate metals (Ag, Au, Pt).

## 4. Conclusions

We conducted a study on the impact of surface treatment for detecting urea on EGTs with different gate electrodes. Using AuNPs, we confirmed the effective immobilization of urease on Ag, Au, and Pt gate metals using APTES and GA, 11-MUA, and 3-MPA. Upon testing the signal responses for urea on the three metals with the three surface treatment methods, we found that APTES and GA consistently exhibited superior detection performance, irrespective of the metal used. This was attributed to the acidic nature of 11-MUA and 3-MPA, which reduced urease activity. Furthermore, it is known that not only urease but all enzymes have their optimal pH range; we anticipate that the same principles can be applied to other enzyme-based sensors.

## Figures and Tables

**Figure 1 micromachines-15-00621-f001:**
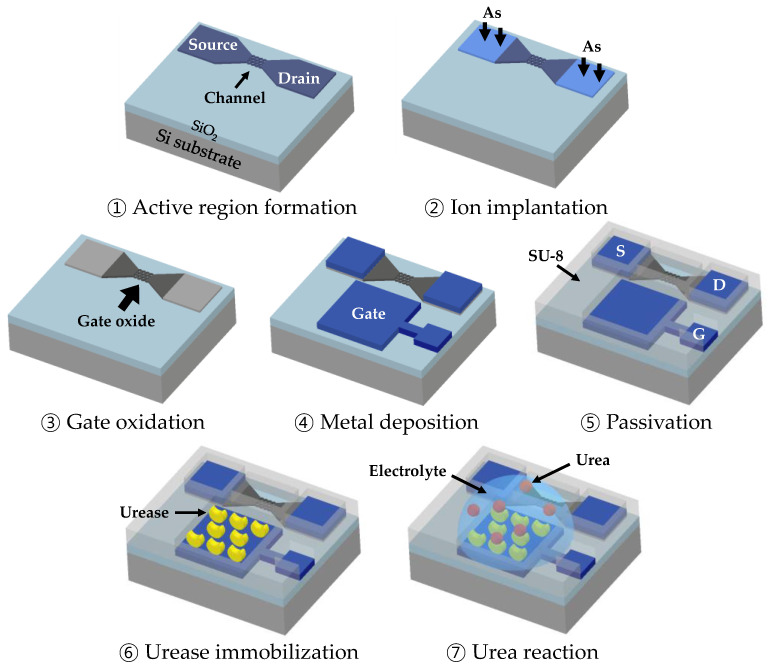
Fabrication and characterization steps of the Si-based electrolyte-gated transistor (EGT).

**Figure 2 micromachines-15-00621-f002:**
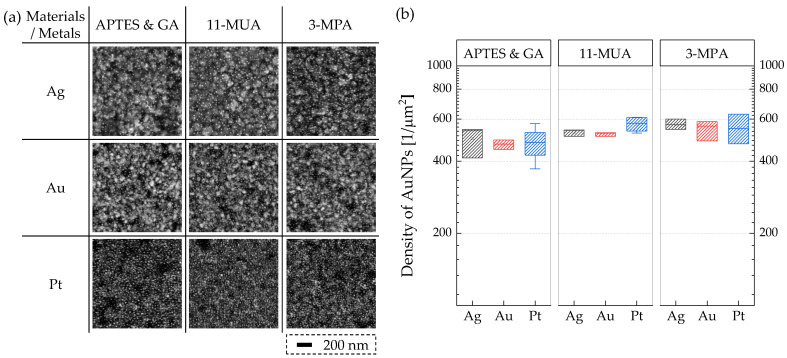
(**a**) Scanning electron microscopy (SEM) images and (**b**) the density of AuNPs immobilized on Ag, Au, and Pt metals for different surface treatments.

**Figure 3 micromachines-15-00621-f003:**
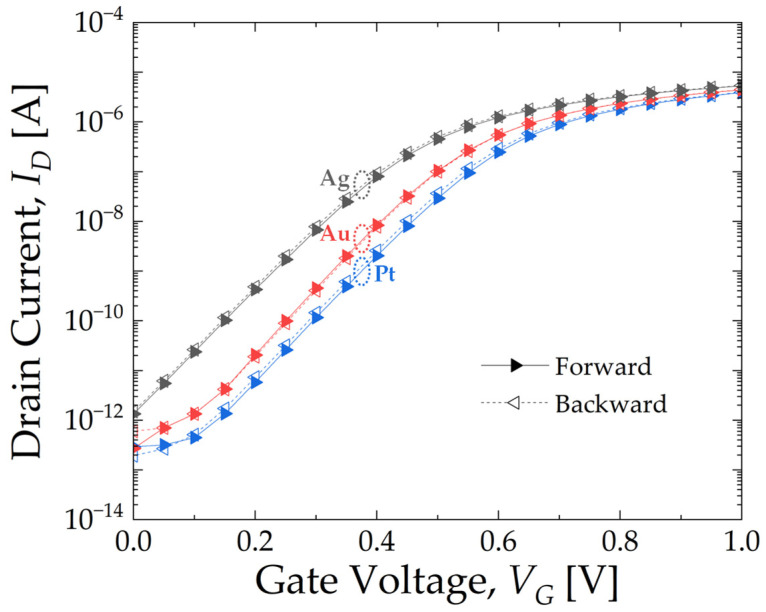
*I_D_*–*V_G_* characteristics of fabricated electrolyte-gated transistors (EGTs) with Ag, Au, and Pt gate electrodes for forward and backward sweeps at *V_D_* = 0.1 V in 1 × PBS solution.

**Figure 4 micromachines-15-00621-f004:**
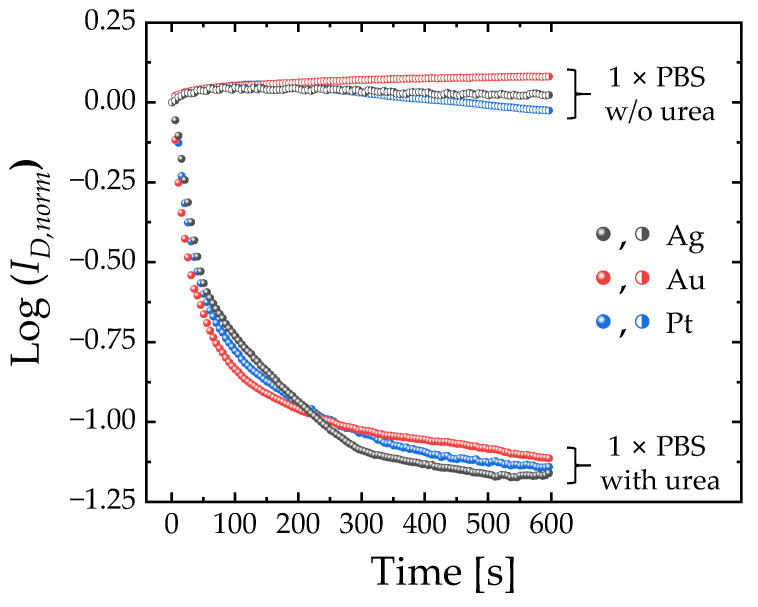
Real-time responses for detecting urea (pUrea 0.5) in urease-functionalized electrolyte-gated transistors (EGTs) using 3-amino-propyl) triethoxysilane (APTES) and glutaraldehyde (GA). The normalized drain current (*I_D, norm_*) was characterized at a fixed *V_G_* of 0.3 V and *V_D_* of 0.1 V. *I_D, norm_* represents the ratio of *I_D_* to *I_D_*_0_, where *I_D0_* denotes the initial *I_D_* measured at t = 0 s.

**Figure 5 micromachines-15-00621-f005:**
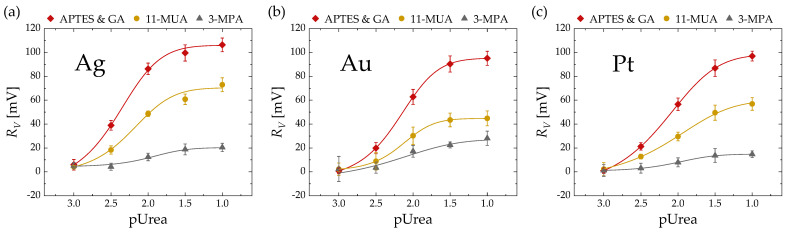
*R_V_* vs. pUrea based on three surface treatment methods (APTES & GA, 11-MUA, 3-MPA) with EGTs featuring (**a**) Ag, (**b**) Au, and (**c**) Pt gate metals. The solid curves represent logistic fitted lines.

**Figure 6 micromachines-15-00621-f006:**
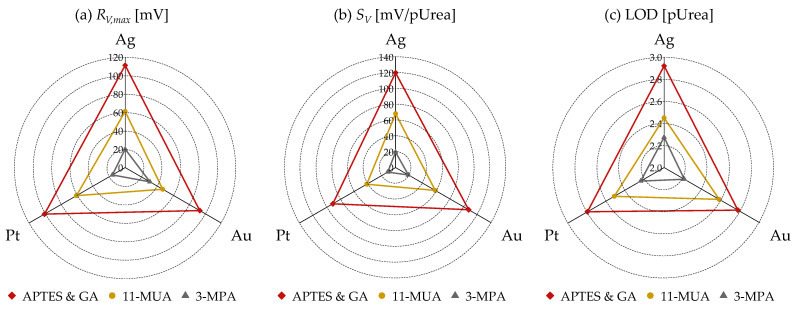
(**a**) *R_V,max_*, (**b**) *S_V_*, and (**c**) limit of detection (LOD) based on three surface treatment methods (APTES and GA, 11-MUA, 3-MPA) and three gate metals (Ag, Au, Pt) extracted from *R_V_* vs. pUrea relationship in Figure 5.

**Figure 7 micromachines-15-00621-f007:**
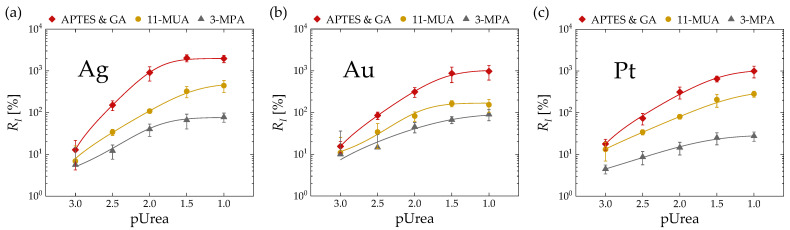
*R_I_* vs. pUrea based on three surface treatment methods (APTES and GA, 11-MUA, 3-MPA) with electrolyte-gated transistors (EGTs) featuring (**a**) Ag, (**b**) Au, and (**c**) Pt gate metals. The solid curves represent logistic-fitted lines.

**Figure 8 micromachines-15-00621-f008:**
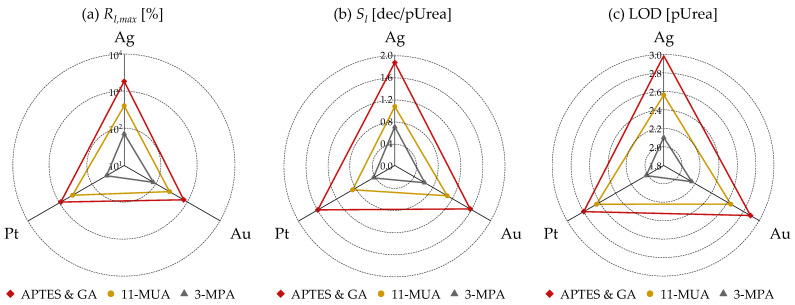
(**a**) *R_I,max_*, (**b**) *S_I_*, and (**c**) limit of detection (LOD) based on three surface treatment methods (APTES and GA, 11-MUA, 3-MPA) and three gate metals (Ag, Au, Pt) extracted from *R_I_* vs. pUrea relationship in Figure 7.

**Figure 9 micromachines-15-00621-f009:**
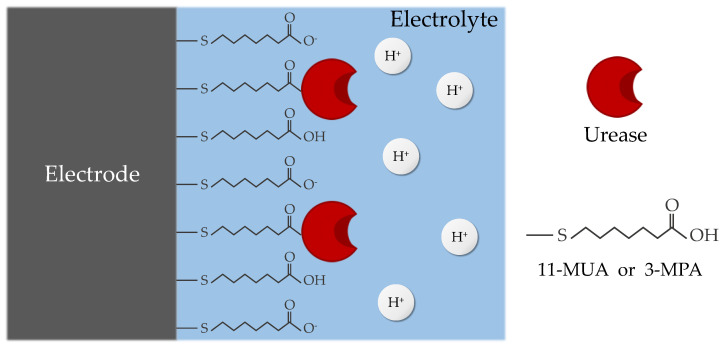
Schematic image of urease immobilized on the gate electrode using 11-MUA or 3-MPA. The carboxyl terminals decompose H^+^ ions, thereby lowering the pH of the surrounding electrolyte.

**Figure 10 micromachines-15-00621-f010:**
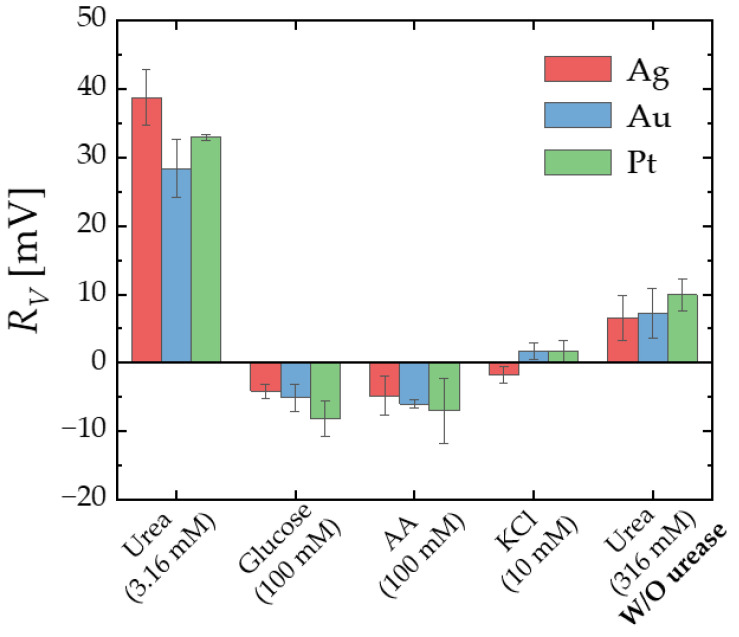
Control experiments: *R_V_* for urea (3.16 mM, pUrea 2.5), glucose (100 mM), AA (100 μM), and KCl (10 mM) with urease functionalized electrolyte-gated transistors (EGTs), and *R_V_* for urea (316 mM, pUrea 0.5) with unmodified EGTs.

**Table 1 micromachines-15-00621-t001:** Materials employed for urease immobilization and their characteristics.

Material	Functional Group		Length	pKa
	Binding with Gate	Binding with Receptor		
APTES and GA [19,20,21,22]	Ethoxy group (OCH_2_CH_3_–)	Amine group (NH_2_–)	1.5 nm	10
11-MUA [23,24,25,26,27]	Thiol group (SH–)	Carboxyl group (COOH–)	1.6 nm	6.5
3-MPA [28,29]	Thiol group (SH–)	Carboxyl group (COOH–)	0.6 nm	5.2

## Data Availability

The data presented in this study are not available due to privacy.

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
