# Peer review of "Influence of Surface Treatments on Urea Detection Using Si Electrolyte-Gated Transistors with Different Gate Electrodes"

_micromachines, 2024, doi:10.3390/mi15050621_

Round 1
Reviewer 1 Report
Comments and Suggestions for Authors
The authors presented a study on transistor applications in biosensing. The approach is systematic and results encouraging. Below are some minor issues:
Abbreviations for surface treatment methods in abstract and introduction should be accompanied with full names.
Transistor testing details should be after its fabrication in the experimental section.
line 90, SU-8 passivation layer may need additional explanations
Multiple scans of IV curves for each sample should be shown for statistical purposes. Any error bars or standard deviations or sampling sizes info?
Any discussion on Hysteresis ? Any reasoning for the observed differences with Au/Ag/Pt? and for the 3 surface treatments?
Comments on the Quality of English Languagegenerally good
Author Response
Please see the attatchment

Reviewer 2 Report
Comments and Suggestions for Authors
In this manuscript, the authors have investigated the impact of surface treatments on Si-based electrolyte-gated transistors (EGTs) for detecting urea. The authors found that the EGTs exhibited exceptional intrinsic electrical properties, including a low subthreshold swing of 80 mV/dec, a high on/off current ratio of 10^6, and negligible hysteresis. Three surface treatment methods (APTES & GA, 11-MUA, 3-MPA) were individually applied to the EGTs with different gate electrodes (Ag, Au, Pt). Gold nanoparticle binding tests were performed to validate the surface functionalization. The topic is very interesting for understanding the EGT and the application. However, several questions should be addressed before this paper can be considered to publish in micromachines.
1. Although the authors used the SEM technique to evaluate the density of AuNPs immobilized on Ag, Au, and Pt metals for different surface treatments,the TEM technique is still needed in order to more accurately characterize this density.
2. In Fig. 3, the authors have shown the transfer curve of fabricated EGTs with Ag, Au, and Pt gate electrodes for forward and backward sweeps, however, why doesn't the author show the value of the negative voltage? More important, according to the results in Fig. 3, the threshold voltage drift is severe, which would seriously affect the reliability of the devices. On the other hands, for forward and backward sweeps in Fig.3, the device characteristics remain essentially the same, why did the author do this treatment.
3. In the manuscript, the authors have selected three metal electrodes (Ag, Au, Pt),does it have the same effect if another electrode chosen, such TiN, Mo, etc.?
4. Contact resistance is very important for device characterization [Analysis of the contact resistance in amorphous InGaZnO thin film Transistors], has the author considered contact resistance?
Comments on the Quality of English LanguageIn this manuscript, the authors have investigated the impact of surface treatments on Si-based electrolyte-gated transistors (EGTs) for detecting urea. The authors found that the EGTs exhibited exceptional intrinsic electrical properties, including a low subthreshold swing of 80 mV/dec, a high on/off current ratio of 10^6, and negligible hysteresis. Three surface treatment methods (APTES & GA, 11-MUA, 3-MPA) were individually applied to the EGTs with different gate electrodes (Ag, Au, Pt). Gold nanoparticle binding tests were performed to validate the surface functionalization. The topic is very interesting for understanding the EGT and the application. However, several questions should be addressed before this paper can be considered to publish in micromachines.
1. Although the authors used the SEM technique to evaluate the density of AuNPs immobilized on Ag, Au, and Pt metals for different surface treatments,the TEM technique is still needed in order to more accurately characterize this density.
2. In Fig. 3, the authors have shown the transfer curve of fabricated EGTs with Ag, Au, and Pt gate electrodes for forward and backward sweeps, however, why doesn't the author show the value of the negative voltage? More important, according to the results in Fig. 3, the threshold voltage drift is severe, which would seriously affect the reliability of the devices. On the other hands, for forward and backward sweeps in Fig.3, the device characteristics remain essentially the same, why did the author do this treatment.
3. In the manuscript, the authors have selected three metal electrodes (Ag, Au, Pt),does it have the same effect if another electrode chosen, such TiN, Mo, etc.?
4. Contact resistance is very important for device characterization [Analysis of the contact resistance in amorphous InGaZnO thin film Transistors], has the author considered contact resistance?
